



# Evaluation of downward and upward solar irradiances simulated by the Integrated Forecasting System of ECMWF using airborne observations above Arctic low-level clouds

Hanno Müller[1], André Ehrlich[1], Evelyn Jäkel[1], Johannes Röttenbacher[1], Benjamin Kirbus[1], Michael Schäfer[1], Robin Hogan[2,3], and Manfred Wendisch[1]

[1]Leipzig Institute for Meteorology (LIM), Leipzig University, Leipzig, Germany
[2]European Centre for Medium-Range Weather Forecasts, Reading, United Kingdom
[3]Department of Meteorology, University of Reading, Reading, United Kingdom

**Correspondence:** Hanno Müller (hanno.mueller@uni-leipzig.de)

**Abstract.** The simulations of upward and downward irradiances by the Integrated Forecasting System (IFS) of the European Centre for Medium-Range Weather Forecasts are compared to broadband solar irradiance measurements from the Arctic CLoud Observations Using airborne measurements during polar Day (ACLOUD) campaign. For this purpose, offline radiative transfer simulations with the ecRad radiation scheme using the operational IFS output were performed. The simulations of the downward solar irradiance agree within the measurement uncertainty. However, the IFS underestimates the reflected solar irradiances above sea ice significantly by $-35\,\mathrm{W\,m^{-2}}$. Above open ocean, the agreement is closer with an overestimation of $29\,\mathrm{W\,m^{-2}}$. A sensitivity study using measured surface and cloud properties is performed with ecRad to quantify the contributions of the surface albedo, cloud fraction, ice and liquid water path and cloud droplet number concentration to the observed bias. It shows that the IFS sea ice albedo climatology underestimates the observed sea ice albedo, causing more than $50\,\%$ of the bias. Considering the higher variability of in situ observations in the parameterization of the cloud droplet number concentration leads to a smaller bias of $-27\,\mathrm{W\,m^{-2}}$ above sea ice and a larger bias of $48\,\mathrm{W\,m^{-2}}$ above open ocean by increasing the range from $36\text{-}69\,\mathrm{cm^{-3}}$ to $36\text{-}200\,\mathrm{cm^{-3}}$. Above sea ice, realistic surface albedos, cloud droplet number concentrations and liquid water paths contribute most to a bias improvement. Above open ocean, realistic cloud fractions and liquid water paths are most important to reduce the model-observation differences.

## 1 Introduction

The Arctic climate has changed more rapidly than the rest of the globe during the recent decades. One clear sign is the reduction of the sea ice extent of the Arctic ocean, particularly in September each year (Serreze and Meier, 2019). Another indicator is the increase of the near-surface air temperature that is in the Arctic more than twice as large as for the whole globe (Rantanen et al., 2022; Wendisch et al., 2023). The ongoing changes of the Arctic climate system emphasize a need for adaptions of forecast models to Arctic-specific peculiarities (Jung et al., 2016). Improved prediction systems for the Arctic would not only be a direct benefit for the future Arctic with its allowed shipping routes (Smith and Stephenson, 2013) but also an indirect benefit



for the forecast in Northern Hemisphere midlatitudes at longer lead times. This is caused by the linkage between the Arctic and midlatitudes that was investigated by e.g. Jung et al. (2014), Cohen et al. (2014), Overland et al. (2015) and Lawrence et al. (2019).

In the Arctic, Numerical Weather Prediction (NWP) models, such as the Integrated Forecasting System (IFS) by the European Centre for Medium-Range Weather Forecasts (ECMWF), often appear more uncertain compared to other regions on the globe (Bauer et al., 2016). The reasons for the lower predictive skills in the Arctic are various and often linked to the particularities of the Arctic climate system. One obvious issue in the Arctic results from the sparse observational coverage, which limits the data assimilation (Bauer et al., 2016; Jung and Matsueda, 2016; Lawrence et al., 2019; Ortega et al., 2022). Furthermore,

the modeling of the sea ice cover is a major obstacle for correctly representing the Arctic surface energy budget but is still uncertain due to the complexity of sea ice dynamics (Day et al., 2022). The representation of low-level Arctic clouds and especially mixed-phase clouds has been identified as another major source of uncertainty (Forbes and Ahlgrimm, 2014). As shown by Morrison and Pinto (2006), especially the cloud microphysical schemes cause uncertainties in the cloud phase and precipitation.

Low-level clouds occur frequently in the Arctic (e.g. Eastman and Warren, 2010; Mioche et al., 2015) and show a pronounced longevity above sea ice and the open ocean (Shupe et al., 2006; Verlinde et al., 2007). Their radiative properties are controlled by a complex system of coupled microphysical and dynamical processes that may differ depending on the surface conditions (Morrison et al., 2012; Wendisch et al., 2019). Especially for optical thin clouds, with a liquid water path (LWP) less than $30\,\mathrm{g\,m^{-2}}$, the cloud radiative effect changes significantly for only small changes of cloud properties (Shupe and Intrieri, 2004).

Thus, these clouds potentially introduce major model uncertainties. To constrain the effect of Arctic low-level clouds on the atmospheric radiation budget, it is necessary to identify the shortcomings of microphysical parameterizations in NWP models to properly predict snow rate and cloud properties (Solomon et al., 2009). A substantial underprediction of cloud LWP together with an overprediction of cloud ice water path (IWP) was revealed by Solomon et al. (2009) and indicated an unrealistic growth of ice particles in the Weather Research Forecast model. Solomon et al. (2023) showed that contemporary models

have difficulties to represent the radiative impact of Arctic clouds and still struggle to keep liquid water at low temperatures. Additionally, the change of the surface type (sea ice or open ocean) when clouds move on or off the sea ice initiates air mass transformations and changes of the cloud dynamics. This transition can result in cloud formation or cloud dissipation and is still poorly represented in NWP models (Pithan et al., 2018; Wendisch et al., 2021).

    Two different concepts to evaluate clouds and their radiative effects in NWP have been applied in the past. The first approach

applies model inter-comparisons (e.g. Klocke and Rodwell, 2014) to identify model uncertainties. This technique is not able to quantify potential model biases compared to reality. The second approach makes use of observed cloud properties. This approach was applied in the past decades to evaluate the representation of clouds in global NWP models using ground-based long-term cloud observations within the framework of Atmospheric Radiation Measurement (ARM) sites (e.g. Yang et al., 2006; Morcrette et al., 2012) or within the Cloudnet (Illingworth et al., 2007) framework (e.g. Hogan et al., 2009; Sinclair

et al., 2022). Due to the nature of observations at a fixed location, only a few of these studies target specific Arctic sites (e.g. Klein et al., 2009; Morrison et al., 2009; Zhao and Wang, 2010; Forbes and Ahlgrimm, 2014). For an evaluation of



the cloud representation in the central Arctic over the Arctic ocean, satellite observations can be utilized, which provide a spatially broader view and come in the polar regions together with frequent overpasses by polar-orbiting satellites, but cause difficulties in the data assimilation of microwave soundings above sea ice (Lawrence et al., 2019). Compared to the long-term
observations of ARM, Cloudnet and satellites, shipborne observations provide short-term observations covering slowly varying different locations. Airborne observations bridge the gap between the ground-based or shipborne observations and the satellite observations and can provide in situ observations of cloud particle properties.

The efforts to improve model representations of Arctic clouds conducted in the past decades covered diverse aspects and quantities. The parameterization and representation of the sea ice albedo in various models was evaluated by e.g. Liu et al.
(2007) and Karlsson and Svensson (2013), who identified the model sea ice albedo to determine both the sign and the amount of its cloud radiative effect. Low-level cloud fractions were assessed in reanalyses by Walsh et al. (2009) and found to be underestimated in summer, which leads to a bias in the solar radiation flux, while Sotiropoulou et al. (2016) evaluated the improvement of the representation of the vertical structure of mixed-phase clouds in the IFS by the change from a diagnostic to a progonostic parameterization of mixed-phase clouds. Integrated microphysical quantities like LWP and IWP were investigated
by Gu et al. (2021) who evaluated these quantities in Arctic reanalyses and found a mean underestimation of both LWP and IWP over the Arctic region compared to satellite observations. The representation of cloud droplet number concentrations in different models was evaluated by Geoffroy et al. (2010), Brenguier et al. (2011) and McCusker et al. (2023), who showed a slight improvement of the overestimation of the liquid cloud mass mixing ratio in low-levels clouds in the Met Office Unified Model (UM) by using representative cloud droplet number concentrations. Stevens et al. (2018) concluded from their model-
intercomparison of cloud condensation nuclei-limited tenuous Arctic clouds that an appropriate treatment of the cloud droplet size distribution within models is important to account for aerosol-cloud interactions. Regarding the IFS, Beesley et al. (2000) evaluated the ECMWF model with observations collected during the Surface Heat Budget of the Arctic Ocean (SHEBA) campaign (Uttal et al., 2002) and identified a much larger observed fraction of liquid water clouds. Tjernström et al. (2021) evaluated the IFS with observations from the Arctic Ocean 2018 (AO2018) expedition (Vüllers et al., 2021) and revealed too
high (near-)surface air temperatures in the IFS. McCusker et al. (2023) evaluated clouds during AO2018 within the IFS that overestimated cloud occurrence below $3\,\mathrm{km}$. Forbes and Ahlgrimm (2014) revealed an underestimation of IFS cloud top albedo compared to observations from the Clouds and the Earth's Radiant Energy System (CERES) project. The bias is linked to an underestimation of liquid water content (LWC) near cloud tops, which results from the parameterization of the cloud phase based on the diagnostic air temperature.

However, these evaluations are often based on remote sensing products, which themself include major uncertainties mostly resulting from several assumptions in the retrieval algorithm, e.g. viewing geometries, instrument sensitivity or the ice crystal shape (Wendisch, 2005). Therefore, Formenti and Wendisch (2008) recommended to compare NWP models in the observational space of radiation, e.g. solar and thermal infrared radiation, radar and lidar reflectivites. Huang et al. (2017) used this approach to evaluate different global reanalyses like ECMWF Reanalysis - Interim and Climate Forecast System Reanalysis
by the National Centers for Environmental Prediction, Matsui et al. (2014) and Berry et al. (2019) to evaluate Earth system models. Observations of airborne solar spectral irradiance have been used by Wolf et al. (2020) in combination with along-track





radiative transfer simulations of the operational ecRad radiation scheme of ECMWF and a benchmark radiative transfer model. Their analysis indicated that IFS underestimates the IWC in a frontal cloud system close to Iceland and that differences in the absorbing spectral band indicate deficiencies in the ecRad ice crystal optical properties. For the Arctic CLoud Observations

Using airborne measurements during polar Day campaign (ACLOUD; Wendisch et al., 2019), Kretzschmar et al. (2020) applied similar measurements and found a pronounced underestimation of the negative cloud radiative effect in the ICON model. This bias was traced back to the cloud condensation nuclei activation in the microphysical scheme. For a specific cloud case observed during ACLOUD, Ruiz-Donoso et al. (2020) investigated the thermodynamic phase of mixed-phase clouds as modeled by ICON large-eddy simulations and found that measured spectral radiances reveal an underestimation of the modeled

ice crystal number concentration. Jäkel et al. (2019b) used ACLOUD observations to analyze the performance of the sea ice albedo scheme used in a regional coupled climate model and found an underestimation of the variability of the sea ice albedo caused by a biased surface albedo parameterization dependence on surface temperature. So far, the comprehensive ACLOUD data set has not been used for any IFS evaluation. While efforts have been made to include sea ice dynamics in IFS to tackle the high sea ice variability close to the sea ice edge (Keeley and Mogensen, 2018), the sea ice albedo in IFS is still based on

climatological values with shortcomings identified by Pohl et al. (2020) using satellite observations.

In this paper, airborne radiation data from the ACLOUD campaign are used to evaluate the representation of Arctic low-level clouds and sea ice albedo in the IFS. The comparison is based on the observational space of solar irradiance and additionally implements active cloud remote sensing and in situ cloud microphysical observations. Section 2 describes the comparison strategy of observations and radiative transfer simulations, which are analyzed in Sect. 3. Based on the additional measurements,

a sensitivity study is presented in Sect. 4. It aims at reducing the differenes between modeled and observed irradiances by improving the surface albedo, cloud fraction and microphysical properties (LWP, IWP and cloud droplet number concentration) with actually measured data. The contributions of each individual parameter to the overall model uncertainty are summarized in the conclusions in Sect. 5.

## 2 Data and methods

### 2.1 Airborne observations

Airborne observations of the ACLOUD campaign (Ehrlich et al., 2019), which took place in May/June 2017 around Svalbard, Norway, provide a comprehensive data set of in situ and remote sensing observations for model evaluation and have been used in this analysis. Two upward and downward looking CMP22 pyranometers (spectral range of 300 nm to 3000 nm) aboard the Polar 5 aircraft (Wesche et al., 2016) measured the broadband solar irradiance (Stapf et al., 2019), which is referred to as

solar irradiance in this study. The uncertainty of the CMP22 irradiance is typically about 2 % according to characterizations by Vuilleumier et al. (2014) for ground-based operations. However, airborne operation of the CMP22 in Arctic conditions may increase these uncertainties depending on solar zenith angle and environmental conditions, as described by Ehrlich et al. (2023) on a high altitude aircraft and Su et al. (2008) in a laboratory study and on active stabilization performance (Wendisch et al., 2001). The data are corrected for the aircraft specific operation as summarized in Ehrlich et al. (2019) including corrections for



aircraft attitude and instrument inertia. For the conditions during ACLOUD a maximum uncertainty of 3 % in regular straight flight sections is assumed.

Spectral solar irradiances are measured on Polar 5 with the Spectral Modular Airborne Radiation measurement sysTem (SMART; Jäkel et al., 2019a; Wendisch et al., 2001) covering the spectral range between 345 nm and 2150 nm with a spectral resolution of 3-15 nm (Ehrlich et al., 2019). Additional remote sensing observations aboard Polar 5 include the Airborne Mobile Aerosol Lidar (AMALi) system (Stachlewska et al., 2010) and the Microwave Radar/radiometer for Arctic Clouds (MiRAC; Kliesch and Mech, 2019; Mech et al., 2019). AMALi is sensitive to both liquid and ice layers, while MiRAC is particularly sensitive to ice layers in the clouds. The cloud top altitude is derived from AMALi using a robust backscatter gradient approach that is retrieval-independent and is only based on the chosen instrument threshold (Kulla et al., 2021). For MiRAC a threshold of $-30\,\mathrm{dBZ}$ equivalent reflectivity factor is applied for identifying cloud particles above the height of 150 m. Altitudes below were not considered to exclude ground clutter. The uncertainty of the radar detector is given by Mech et al. (2019) as 0.5 dBZ, which only slightly affects the uncertainty in detecting cloud layers.

The Polar 6 aircraft was equipped with numerous in situ cloud probes. This study makes use of cloud particle number concentrations measured by the Small Ice Detector Mark 3 (SID-3; Schnaiter and Järvinen, 2019; Hirst et al., 2001; Vochezer et al., 2016). To characterize the surface conditions, sea ice concentration satellite data are retrieved from the Advanced Microwave Scanning Radiometer 2 (AMSR2) measurements (Melsheimer and Spreen, 2019; Spreen et al., 2008).

## 2.2 Radiative transfer simulations

### 2.2.1 Integrated Forecasting System

The results of the simulations presented in this paper have been achieved with the 'Atmospheric Model high resolution' configuration (HRES) of the IFS of the ECMWF. Model cycle 43r1 was operational during the time of ACLOUD. A detailed description of the IFS can be found at https://www.ecmwf.int/en/publications/ifs-documentation. To evaluate the short term forecast and not the model initialisation, the 00 UTC runs with hourly forecast steps are used, issued about 12 hours before each flight. The prognostic variables from the IFS are available at 137 model levels. About 32 levels lay below a typical flight altitude of 3000 m with the highest vertical resolution of about 30 m close to the ground. The spacing between grid points of the longitude-latitude-grid is 0.07° both along the longitude and the latitude axis, resulting in a horizontal resolution of 1.4 km to 7.8 km in the campaign region. The surface type is classified as open ocean/sea ice when both the sea ice concentrations by AMSR2 and the IFS are below 0.01 %/above 60 %.

### 2.2.2 ecRad radiation scheme

The prognostic variables air pressure, air and skin temperature, specific humidity and cloud fraction from the IFS serve as direct input to the ecRad radiation scheme (Hogan and Bozzo, 2018). The ecRad version 1.4.0 is applied in an offline mode, which allows to run sensitivity studies. In addition, the required quantities liquid/ice cloud mass mixing ratios are calculated as sums of the specific cloud liquid/ice water content and the specific rain/snow water content. Similarly, the effective radii are



no prognostic variables in the IFS and need to be calculated consistently to the IFS. The definition of ice cloud effective radius follows the parameterization by Sun and Rikus (1999) and Sun (2001). The definition of liquid cloud effective radius in the IFS is based on the parameterization by Martin et al. (1994) with an adjustment by Wood (2000).

The IFS distinguishes over open ocean between a spectrally constant surface albedo value of 0.06 for diffuse radiation and a solar zenith angle dependent surface albedo given by Taylor et al. (1996) for direct radiation. Here, the open ocean albedo is approximated with the diffuse albedo only. The surface albedo used in ecRad is composed of this open ocean albedo and the sea ice albedo, which is based on the one-dimensional sea ice model by Ebert and Curry (1993) providing a monthly mean climatology of the spectral surface albedo in six solar bands (boundaries at $0.185, 0.250, 0.440, 0.690, 1.190, 2.38$ and $4.0\,\mu\text{m}$).

This climatology is interpolated to the day of the specific flight. The surface type composition is obtained from the prognostic sea ice cover in the IFS.

Mass mixing ratios of $CH_4$, $CO$, $NO_2$ and eleven different hydrophilic and hydrophobic aerosol species from the Copernicus Atmosphere Monitoring Service (CAMS) global reanalysis (EAC4) were extracted from the CAMS Atmosphere Data Store (Inness et al., 2019). Similarly, volume mixing ratios of $CO_2$ (Chevallier et al., 2010, 2019) and $N_2O$ (Thompson et al.,

2011) from the CAMS global inversion-optimised greenhouse gas fluxes and concentrations product were included. Ozone concentrations were obtained from operational ozone soundings above Ny-Ålesund, Svalbard. The TOA solar irradiance of $1360.8\,\text{W}\,\text{m}^{-2}$ (Kopp and Lean, 2011) is adjusted to the Earth-Sun-distance from noon of every flight day.

In the operational configuration, ecRad uses the McICA radiative transfer solver (Pincus et al., 2003). However, this solver does not provide spectrally resolved irradiances across the vertical column, which is needed for a direct comparison in flight

altitude. Therefore, the operational solver is replaced by the Tripleclouds solver (Shonk and Hogan, 2008). A comparison of surface irradiances (not shown here) showed that both solvers do not significantly differ. The exponential-random cloud overlap assumption is applied in the Tripleclouds solver. Cloud overlap is parameterized by the overlap decorrelation length, which is calculated after Shonk et al. (2010), Eq. 13. The aerosol scattering properties are based on the IFS version cycle 43r1, in combination with the operational aerosol type classification from cycle 43r3. For the ice crystal optical properties, the

operational parameterization from Fu (1996) and Fu et al. (1998) is chosen. The used gas absorption model is based on the Rapid Radiative Transfer Model for GCMs (RRTM-G; Mlawer et al., 1997) and defines the spectral resolution of ecRad in terms of 14 shortwave bands. Running ecRad in the described configuration provides spectral upward and downward irradiances at the interfaces of the 137 full levels in the 14 shortwave bands, which are then integrated to broadband irradiances.

## 2.3  Considering the scale mismatch

For the comparison between the measurements and the simulations, the aircraft is assumed to artificially fly through the model grid space. For this purpose, the different spatial scales of airborne observations and simulations have to be considered. The mean horizontal grid spacing of the simulations is in the range of 4.6 km. The time Polar 5 needs to fly between two grid points accounts for about $60\,\text{s}$ with an average speed of $80\,\text{m}\,\text{s}^{-1}$. Therefore, the airborne data are averaged over $60\,\text{s}$ and ecRad is run every $60\,\text{s}$ at the mean aircraft location during the corresponding averaging track interval. The ecRad input is extracted from

IFS according to the closest grid box to the mean position of Polar 5 and to the nearest 1 hour IFS time step to the $60\,\text{s}$ interval.



This results in a maximum time offset of 30 minutes between simulation and observation. Temporal interpolation of the IFS output was deliberately omitted to avoid smeared states in ecRad input variables. Similarly, without interpolation, the ecRad output at the closest model level to the flight altitude is selected for the comparison.

The statistical comparison between observations and simulations is done using frequency distributions of solar irradiance. This additionally accounts for spatial and temporal mismatches, which would be present in a point-by-point comparison. The frequency distributions are compared using two quantities. On the one hand, the deviation $\Delta F$ of their mean values is calculated via

$$\Delta F = \bar{P}_{\mathrm{ecRad}} - \bar{P}_{\mathrm{obs}}, \tag{1}$$

where $\bar{P}_{\mathrm{ecRad}}$ and $\bar{P}_{\mathrm{obs}}$ are the means of the number frequency distributions of solar irradiances $P_{\mathrm{ecRad}}$ and $P_{\mathrm{obs}}$ from ecRad and the observations. On the other hand, the Hellinger distance $\mathcal{H}$ (Hellinger, 1909) is used as a metric to include the shape of the frequency distributions in the comparison and is calculated following

$$\mathcal{H}(P_{\mathrm{obs}}, P_{\mathrm{ecRad}}) = \frac{1}{\sqrt{2}} \sqrt{\sum_{i=1}^{n} \left( \sqrt{p_{\mathrm{obs,i}}} - \sqrt{p_{\mathrm{ecRad,i}}} \right)^2}, \tag{2}$$

with $P_{\mathrm{obs}} = (p_{\mathrm{obs,1}}, \ldots, p_{\mathrm{obs,n}})$ and $P_{\mathrm{ecRad}} = (p_{\mathrm{ecRad,1}}, \ldots, p_{\mathrm{ecRad,n}})$. The index identifies the center of each bin. $\mathcal{H}$ ranges from 0 to 1, where a value of 0 corresponds to identical distributions and a value of 1 characterizes fully independent distributions. In the following, both described quantities are accompanied by arrows ($\uparrow, \downarrow$), indicating the upward or downward direction.

## 3  Comparison of simulated and measured solar irradiances

A comparison is carried out between simulated and measured solar irradiances in order to quantify the representation and its uncertainty of Arctic low-level clouds in the IFS. To achieve this, the analysis is limited to scenes when no higher clouds located between the flight level of Polar 5 and top of atmosphere (TOA) are present. This condition needs to hold for both observations and simulations and guarantees, that the reflected upward solar irradiance is only affected by possible clouds below the aircraft and not contaminated by attenuation of the incoming irradiance. Scenes are identified as cloud-free above Polar 5 when the standard deviation of the CMP22 downward solar irradiance within a $60\,\mathrm{s}$ interval does not exceed the mean value by $0.7\,\%$. Cloud-free conditions in the IFS are given, when the sum of the fraction of cloud cover in all model levels above the aircraft flight level is below $0.02$. These thresholds reliably exclude mid-level and cirrus clouds from the analysis. The analysis is further limited to periods when all remote sensing instruments provided data so that the retrieval of cloud fractions and of LWP above open ocean is available. The filtering results in 501 scenes ($60\,\mathrm{s}$ intervals) above sea ice and 210 scenes ($60\,\mathrm{s}$ intervals) above open ocean contributed by nine out of 19 research flights, which are shown as flight tracks in Fig. 1. All scenes lay west of Svalbard with the majority above sea ice with a relatively high sea ice concentration.

Figure 2 shows the frequency distributions of upward and downward solar irradiances measured by the CMP22 pyranometer and simulated by ecRad separated for sea ice and open ocean. The downward irradiances cover the range from $440\,\mathrm{W\,m^{-2}}$ to



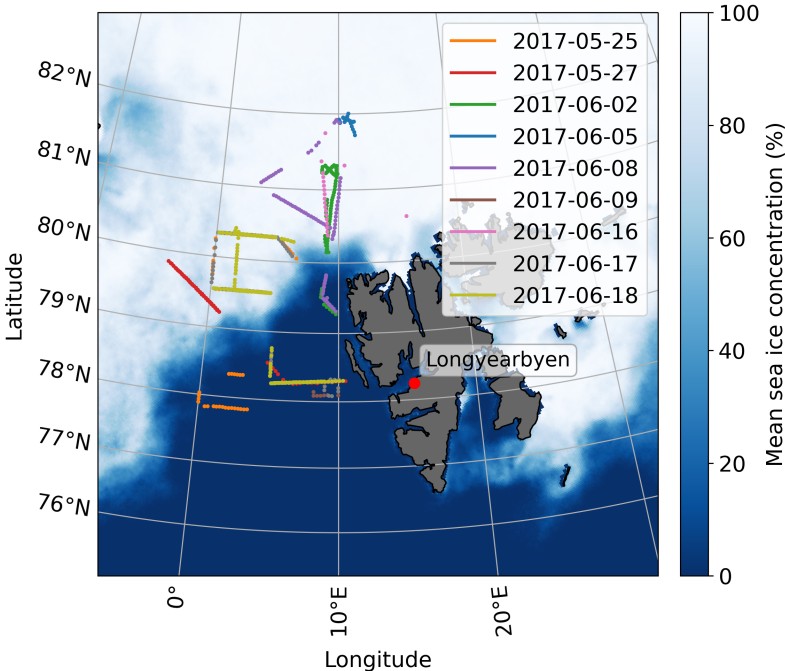

**Figure 1.** Sections of flight tracks of nine research flights that are included in the analysis after filtering. The mean sea ice concentration during the ACLOUD campaign derived from AMSR2 is shown in the background layer.

$670\,\mathrm{W\,m^{-2}}$. The lower irradiances above open ocean result from the larger solar zenith angles during these flight sections in the morning after takeoff and in the afternoon before landing. There is good agreement between the simulated and observed distribution of downward irradiances with $\Delta F^{\downarrow} = -14\,\mathrm{W\,m^{-2}}$ above sea ice (Fig. 2a) and $\Delta F^{\downarrow} = -8\,\mathrm{W\,m^{-2}}$ above open

225  ocean (Fig. 2b), which is within the 3 % maximum uncertainty of the CMP22 measurements. The corresponding $\mathcal{H}^{\downarrow}$ are calculated with 64 bins of $10\,\mathrm{W\,m^{-2}}$ width from $35\,\mathrm{W\,m^{-2}}$ to $665\,\mathrm{W\,m^{-2}}$ and are 0.42 above sea ice and 0.37 above open ocean.

The observations above sea ice surface range between $300\,\mathrm{W\,m^{-2}}$ and mainly $530\,\mathrm{W\,m^{-2}}$ (Fig. 2c). The simulations show a similar amount of low irradiances but end abruptly at $450\,\mathrm{W\,m^{-2}}$. This upper limit in the IFS seems to be limited to clouds

230  over sea ice. While the distribution of upward irradiances above sea ice is relatively narrow due to the high albedo of the sea ice reducing the cloud radiative effect, the distribution of the upward irradiances above the open ocean with its dark surface and low surface albedo is broader (Fig. 2d). It covers a range of irradiances from $150\,\mathrm{W\,m^{-2}}$ to $470\,\mathrm{W\,m^{-2}}$ in the simulations and from $30\,\mathrm{W\,m^{-2}}$ to mainly $510\,\mathrm{W\,m^{-2}}$ in the observations. The low values of the measurements result from scenes without any clouds below the Polar 5 where the dark open ocean absorbs the major part of the incoming solar radiation. High values

235  correspond to cloudy scenes reflecting a large amount of the incoming solar irradiance (Fig. 2d). Over ocean, higher upward irradiances are simulated, despite the lower surface albedo. The means above sea ice show a bias of $\Delta F^{\uparrow} = -35\,\mathrm{W\,m^{-2}}$ with a $\mathcal{H}^{\uparrow}$ of 0.48 and above ocean a bias of $\Delta F^{\uparrow} = 29\,\mathrm{W\,m^{-2}}$ with a $\mathcal{H}^{\uparrow}$ of 0.36.





**Figure 2.** Distribution of (a,b) downward and (c,d) upward solar irradiances for above Polar 5 cloud-free scenes measured by the CMP22 and simulated by ecRad above (a,c) sea ice and (b,d) open ocean. The values in the corner indicate the difference of the mean irradiances between ecRad and CMP22, the corresponding $\mathcal{H}^{\uparrow,\downarrow}$ and the number of included scenes.

While the magnitudes of $\Delta F^{\downarrow}$ are not significant, $\Delta F^{\uparrow}$ exceed the measurement uncertainty and suggest that either surface or cloud properties are not represented correctly in the IFS.

## 4 Sensitivity study

There are numerous possible contributors to the observed bias of the reflected solar irradiance. In principle, the radiative transfer and thus, the reflected solar irradiance is mostly affected by the surface albedo, the cloud fraction and the optical depth of the cloud, neglecting the minor impact of atmospheric gases and aerosols. Following Kokhanovsky (2004), the optical depth





**Table 1.** Overview of performed sets of ecRad simulations indicating which parameter was adjusted by which source.

| ecRad run | adjusted parameter | source |
|---|---|---|
| 1 (reference) | - | IFS cy43r1 |
| 2 | $\alpha$ | SMART |
| 3 | $f_{\text{cloud}}$ | AMALi/MiRAC |
| 4 | IWP | non-observation based |
| 5 | LWP | MiRAC |
| 6 | LWP | non-observation based |
| 7 | $N_{\text{d}}$ | SID-3 |

$\tau$ is related to the cloud properties LWP or LWC, particle effective radius $r_{\text{eff}}$ and density of water $\rho_{\text{w}}$ via

$$\tau = \frac{3}{2} \cdot \frac{1}{\rho_{\text{w}}} \cdot \frac{\text{LWP}}{r_{\text{eff}}} = \frac{3}{2} \cdot \frac{1}{\rho_{\text{w}}} \cdot \frac{\int_z \text{LWC} \, dz}{r_{\text{eff}}}, \tag{3}$$

with a vertical integration over the altitude $z$. However, the optical depth is neither a direct user variable in IFS nor in ecRad. According to the IFS documentation, the mean liquid effective radius $r_{\text{eff}}$ is parameterized following a variation from Martin et al. (1994) by

$$r_{\text{eff}} = \left( \frac{3 \, E_{\text{d}} \, (\text{LWC} + \text{RWC})}{4 \, \pi \, \rho_{\text{w}} \, k \, N_{\text{d}}} \right)^{\frac{1}{3}}, \tag{4}$$

where $E_{\text{d}}$ is an enhancement factor considering an increased dispersion of the droplet size spectrum (Wood, 2000), LWC and RWC are the liquid and rain water content, $k$ is a factor depending on the relative dispersion of the cloud droplet spectrum set to 0.77 above ocean and $N_{\text{d}}$ is the cloud droplet number concentration. $N_{\text{d}}$ is parameterized via the aerosol number and mass concentrations as a function of prognostic 10-m wind speed accounting for the injection of sea spray aerosols from the ocean (Martin et al., 1994; Boucher and Lohmann, 1995; Lowenthal et al., 2004; Erickson et al., 1986; Genthon, 1992).

These dependencies of the cloud radiative properties and the reflected irradiance finally suggest a sensitivity study testing the contribution of the individual parameters to the observed bias $\Delta F^{\uparrow}$. For the sensitivity runs, the IFS input to ecRad is adjusted for surface albedo, cloud fraction, LWP, IWP and $N_{\text{d}}$ individually based on observations where possible. The source of the observed parameters are listed in Table 1 and described below. The reference case is identical to the simulations shown in Sect. 3, where the operational IFS output is fed to ecRad to simulate the solar irradiances.

## 4.1 Sea ice albedo

In the area covered by the ACLOUD campaign the surface albedo conditions were rather constant above open ocean but more variable above sea ice, which was affected by the melt season. Therefore, this sensitivity run is limited to the observations over sea ice. A realistic constraint of the sea ice albedo is deduced from SMART albedometer measurements from low-level flight





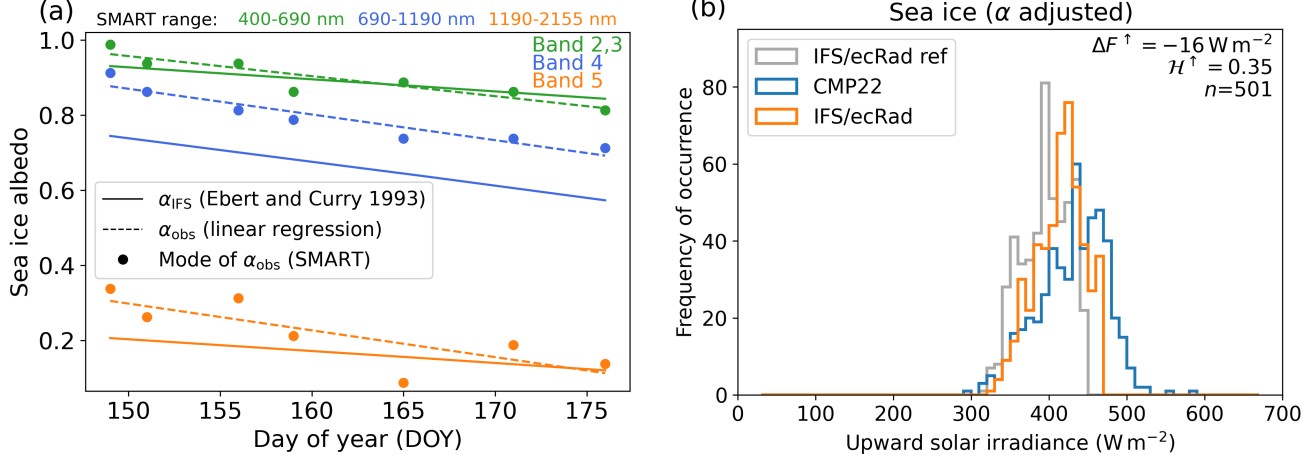

**Figure 3.** (a) Time series of modes of measured sea ice albedo in the 400-690 nm, 690-1190 nm and 1190-2155 nm band and IFS sea ice albedo climatology. The dashed lines show the parameterization for the measurements. (b) Distribution of upward solar irradiances for above Polar 5 cloud-free scenes above sea ice measured by the CMP22 (blue) and simulated by ecRad with the adjusted sea ice albedo (orange) together with the reference simulations (grey).

sections. Wavelength ranges from 400-690 nm, 690-1190 nm and 1190-2380 nm are chosen for a wavelength band approach.

Figure 3a shows the sea ice albedos from measurements of all below-cloud flight sections at flight altitudes below 300 m over sea ice ($\alpha_{\mathrm{obs}}$, mode values) together with the IFS sea ice albedo climatology in the different bands. The influence of the season is obvious, as the measured sea ice albedo values decrease, mainly because of snow metamorphism to larger grain sizes due to the increase in skin temperature, accumulating liquid water in the snow layer and formation of the surface scattering layer (Rosenburg et al., 2023). The IFS sea ice albedo climatology assumes a slower melting season in bands 2, 3 and 5. It

underestimates the surface albedo in bands 2 and 3 at the beginning and overestimates it at the end of the campaign, while there is an underestimation in bands 4 and 5 during the whole campaign. These findings support the shortcomings identified by Pohl et al. (2020) with climatologically fixed transitions between the dry snow, melting snow and bare sea ice albedo from Ebert and Curry (1993).

  The impact of the faster sea ice albedo reduction and the underestimation of the sea ice albedo on the irradiances is inves-

tigated by adjusting the sea ice albedo climatology in a set of ecRad simulations. Linear regressions of the measured sea ice albedo in the three SMART wavelength ranges are used to estimate $\alpha_{\mathrm{obs}}$ at each flight day. The following adjustments are made to the spectral albedo bands in ecRad:

$$\alpha_{\mathrm{IFS,Band\,2/3}} = \alpha_{\mathrm{IFS,250-440\,nm/440-690\,nm}} = \alpha_{\mathrm{obs,400-690\,nm}} = 1.757 - 0.005 \cdot \mathrm{DOY} \tag{5}$$

$$\alpha_{\mathrm{IFS,Band\,4}} = \alpha_{\mathrm{IFS,690-1190\,nm}} = \alpha_{\mathrm{obs,690-1190\,nm}} = 1.896 - 0.007 \cdot \mathrm{DOY} \tag{6}$$

$$\alpha_{\mathrm{IFS,Band\,5}} = \alpha_{\mathrm{IFS,1190-2380\,nm}} = \alpha_{\mathrm{obs,1190-2155\,nm}} = 1.367 - 0.007 \cdot \mathrm{DOY}, \tag{7}$$





**Table 2.** Differences $\Delta F^\uparrow$ of the mean simulated and observed upward solar irradiance distributions of ecRad simulations and CMP22 observations and their corresponding $\mathcal{H}^\uparrow$ for all sets of simulations.

| ecRad run | | sea ice $\Delta F^\uparrow$ (W m$^{-2}$) | $\mathcal{H}^\uparrow$ | open ocean $\Delta F^\uparrow$ (W m$^{-2}$) | $\mathcal{H}^\uparrow$ |
|---|---|---|---|---|---|
| 1 | (reference) | -35 | 0.48 | 29 | 0.36 |
| 2 | $\alpha$ | -16 | 0.35 | 29 | 0.36 |
| 3 | $f_{\mathrm{cloud}}$ | -35 | 0.48 | 18 | 0.35 |
| 4 | IWP -50 % | -35 | 0.48 | 27 | 0.35 |
| 4 | IWP +50 % | -35 | 0.48 | 30 | 0.38 |
| 5 | LWP$_{\mathrm{obs}}$ | - | - | -28 | 0.42 |
| 6 | LWP -50 % | -45 | 0.54 | -5 | 0.39 |
| 6 | LWP +50 % | -28 | 0.47 | 47 | 0.39 |
| 7 | $N_{\mathrm{d}}$ | -27 | 0.41 | 48 | 0.38 |

where DOY is the day of the year. Bands 2 and 3 are considered together due to the sparse SMART coverage of band 2, bands 1 and 6 are kept unchanged as they lie out of the SMART wavelength range.

The results of the modified ecRad run are shown in Fig. 3b and compared to the reference run and the observations. Due to the higher sea ice albedo especially for band 4, the sensitivity run results on average in higher upward solar irradiances. The

bias $\Delta F^\uparrow$ decreases accordingly from $-35\,\mathrm{W\,m}^{-2}$ to $-16\,\mathrm{W\,m}^{-2}$. The corresponding $\mathcal{H}^\uparrow$ decreases from 0.48 to 0.35. These and all following values from the subsequent ecRad runs are summarized in Table 2. Thus, the replacement of the original sea ice albedo reduces the gap between the simulated and the observed irradiances by more than 50 %. This indicates, that the representation of the sea ice albedo in the IFS causes one major part of the disagreement. Another major part may be caused by the representation of clouds.

## 4.2  Cloud fraction


The cloud fraction of the IFS is compared to airborne remote sensing observations. A lidar-based cloud mask from the AMALi cloud top altitude product (Kulla et al., 2021) is used in combination with a radar-based cloud identification from MiRAC. Merging both types of cloud identification leads to a remote sensing based cloud fraction $f_{\mathrm{cloud,\,RS}}$ that accounts for the different sensitivites of radar and lidar. Separated for sections above sea ice and open ocean, Fig. 4 compares the combinations

of observed and forecasted cloud fractions combined with the corresponding mean upward irradiance differences between ecRad simulations and observations. An ideal representation of the observed clouds in the IFS would entail all data circles to lie on the dashed diagonal line with white color indicating no bias in the observed and simulated solar irradiances. However, especially above open ocean, the remote sensing cloud fraction covers the whole range from cloud-less to overcast conditions, while the IFS shows only little variability with cloud fractions ranging between 60 % and 100 %. The data below the dashed

diagonal correspond to an overestimation of the cloud fraction by the IFS, which causes the positive irradiance differences to





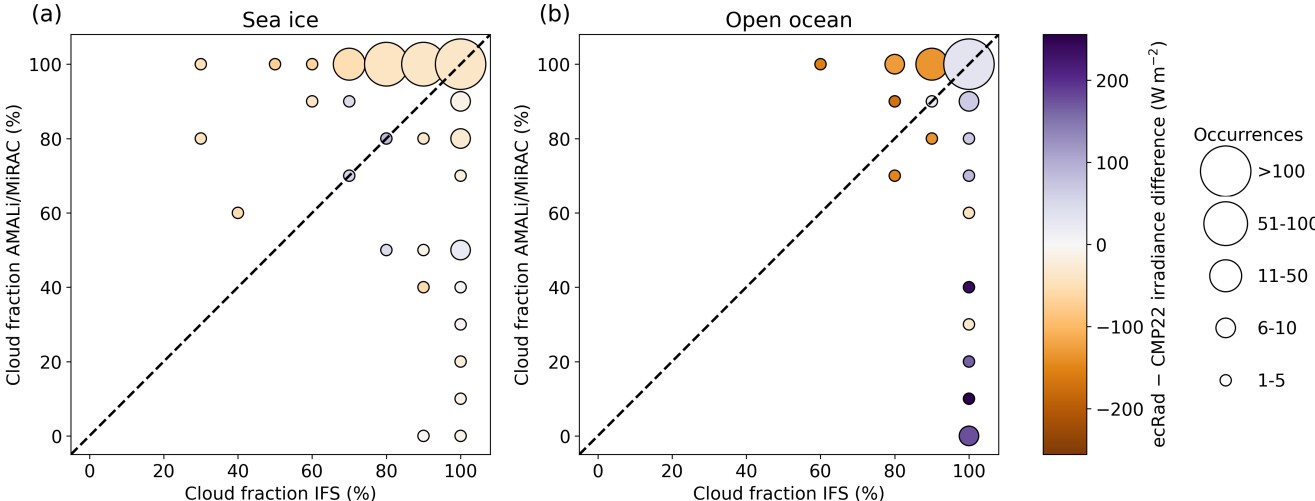

**Figure 4.** Two-dimensional frequency distributions of IFS cloud fractions and observational cloud fractions based on AMALi and MiRAC above (a) sea ice and (b) open ocean with the upward irradiance differences between ecRad simulations and CMP22 observations as circle colors.

dominate. The data above the 1:1 line correspond to an underestimation of the cloud fraction by the IFS and, thus, leads to negative irradiance differences. Data points, where the cloud fractions agree are mostly observed for overcast conditions. In this case, the bias of the ecRad simulations is negative above sea ice and positive above open ocean.

A set of ecRad simulations is performed where the prognostic cloud fraction $f_{\mathrm{cloud, IFS, level}}$ is replaced by the observations

taking into account the vertical distribution of the clouds. As a basic approach, the IFS cloud profiles are kept constant. To account for maximum overlap, this approach ensures that the cloud fraction at the level where it has the profile's maximum cloud fraction is replaced by the remote sensing cloud fraction. All other levels are scaled accordingly adopting the original shape of the cloud fraction profile. This is realized by replacing $f_{\mathrm{cloud, IFS, level}}$ with $f'_{\mathrm{cloud, level}}$ calculated via

$$f'_{\mathrm{cloud, level}} = f_{\mathrm{cloud, RS}} \cdot \frac{f_{\mathrm{cloud, IFS, level}}}{f_{\mathrm{cloud, IFS, max}}}, \qquad (8)$$

where $f_{\mathrm{cloud, IFS, max}}$ is the maximum cloud fraction of all 137 model levels.

Figure 5 compares the irradiance distributions from the ecRad simulations with the replaced cloud fraction to the reference run and the observations. Above sea ice, the simulated irradiance distribution does not change significantly. Due to the high surface albedo small changes of $f_{\mathrm{cloud}}$ do not significantly reduce the reflected radiation. Thus, $\Delta F^{\uparrow}$ remains at $-35\,\mathrm{W\,m^{-2}}$ with $\mathcal{H}^{\uparrow}$ remaining at 0.48. Above open ocean, the replacement by the observed cloud fraction leads to a higher amount of data

with low reflected irradiances. This results from the overestimation of prognostic $f_{\mathrm{cloud}}$ that may be linked to broken cloud conditions that cannot be resolved by the IFS. $\Delta F^{\uparrow}$ is reduced by 37 % from $29\,\mathrm{W\,m^{-2}}$ to $18\,\mathrm{W\,m^{-2}}$ with a corresponding $\mathcal{H}^{\uparrow}$ decrease from 0.36 to 0.35 (see Table 2).





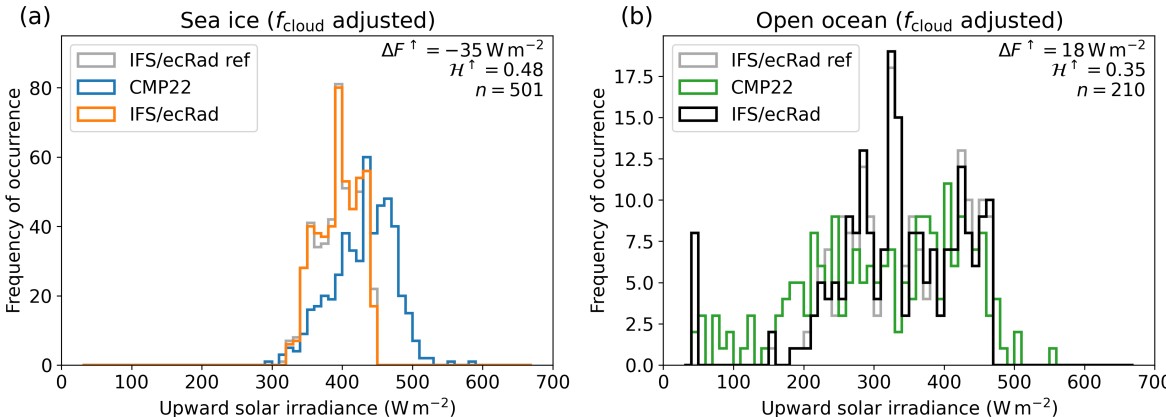

**Figure 5.** Distribution of upward solar irradiances above (a) sea ice measured by the CMP22 (blue) and simulated by ecRad with the adjusted cloud fractions (orange) together with the reference simulations (grey) and above (b) open ocean measured by the CMP22 (green) and simulated by ecRad with the adjusted cloud fractions (black) together with the refence simulations (grey).

## 4.3 Microphysical cloud properties

### 4.3.1 Ice water path

The low-level clouds observed during ACLOUD are mostly of mixed-phase character although dominated by liquid droplets (Ruiz-Donoso et al., 2020; Klingebiel et al., 2023). To test the relevance of the representation of ice crystals in the IFS to the cloud-reflected solar irradiance, no direct observations are available from ACLOUD. Therefore, the prognostic IWP in terms of the specific cloud ice water content is both increased and reduced on a theoretical basis by 50 % in two sets of simulations. Over sea ice, the simulated upward irradiance did not change. Above open ocean, the mean simulated irradiance is only increased

by $1\,\mathrm{W\,m^{-2}}$ when the IWP is increased by 50 % and is only reduced by $2\,\mathrm{W\,m^{-2}}$ when the IWP is reduced by 50 %. Thus, more cloud ice increases the bias of the irradiance simulations and less cloud ice reduces the bias. These small effects confirm the relatively low IWP during ACLOUD reported by Klingebiel et al. (2023) and indicate that the cloud droplets dominate the cloud radiative properties. Here, ice crystals may not directly cause the bias between simulated and observed irradiances.

### 4.3.2 Liquid water path

To adjust the prognostic LWP in the IFS with observations, LWP measurements derived from passive microwave remote sensing observations on Polar 5 are applied. However, the LWP product by the passive 89 GHz channel from MiRAC (Kliesch and Mech, 2019) is only available above open ocean. Above sea ice with its high emissivity the retrieval sensitivity is not sufficient. To confront the observed LWP with the IFS output, the prognostic liquid cloud mass mixing ratio is converted to the LWC and vertically integrated to the LWP below flight altitude.





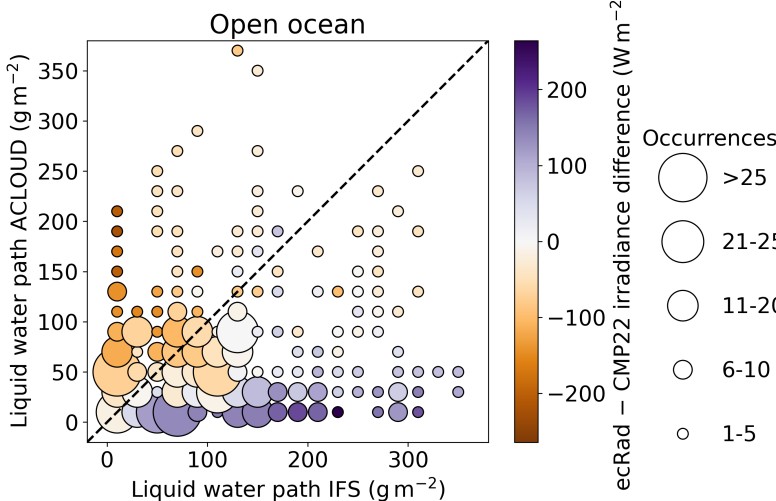

**Figure 6.** Combinations of prognostic IFS LWP and observed LWP based on MiRAC with classes of absolute frequency as circle size and upward solar irradiance differences between ecRad simulations and CMP22 observations as circle colors.

The combinations of the observed and the prognostic LWP are shown in Fig. 6, which reveal the tendency of the IFS to predict too high LWP, while the observations rarely show a LWP above $150 \, \mathrm{g \, m^{-2}}$. This point-by-point mismatch indicates that cloud heterogeneity is high for the observed clouds. This is typical for low-level clouds over open ocean (Schäfer et al., 2018), especially when linked to cold air outbreaks. The exact position of the horizontal cloud structures cannot be forecasted precisely and also may change within the time offset between observations and IFS output. However, as shown in Fig. 6, the differences of irradiance correlate with the mismatch in LWP. Even within one single research flight both overestimations and underestimations of the observed LWP by a factor of two occur, which does not enable a generalization to the scenes above sea ice without a LWP retrieval.

A set of ecRad simulations is performed with adjusted LWP. The specific cloud liquid water content $q_{\mathrm{liq}}$ in ecRad is replaced with

$$q'_{\mathrm{liq}} = q_{\mathrm{liq}} \cdot \frac{\mathrm{LWP_{obs}}}{\mathrm{LWP_{IFS}}}, \tag{9}$$

where $\mathrm{LWP_{obs}}$ is the ACLOUD LWP and $\mathrm{LWP_{IFS}}$ the prognostic integrated LWP, so that the LWC profile shape is kept. The liquid effective radius is recalculated respectively, considering the changed LWC in Eq. 4. After adjusting, $r_{\mathrm{eff}}$ is limited to 4-30 $\mu$m to match the IFS constraints again.

The distributions of the upward solar irradiance for the observations, the reference simulations and the adjusted simulations with replaced LWP and $r_{\mathrm{eff}}$ are shown in Fig. 7. The impact of adjusting the LWP based on observations on the irradiance distributions leads to a change in the correct direction by reducing the upward solar irradiances but the impact is too strong resulting in a conversion of the overestimation to an underestimation. The adjustment overcompensates the reference bias





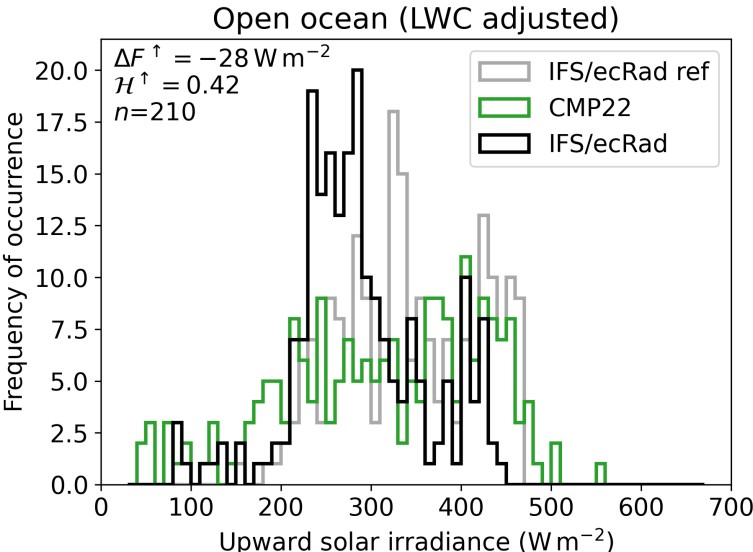

**Figure 7.** Distribution of upward solar irradiances above open ocean measured by the CMP22 (green) and simulated by ecRad (black) with the adjusted LWP and liquid effective radius based on the MiRAC LWP scaling of the prognostic LWP together with the reference simulations (grey).

between the simulated and the observed distribution of upward irradiances with $\Delta F^{\uparrow} = -28\,\mathrm{W\,m^{-2}}$ and the corresponding $\mathcal{H}^{\uparrow}$ increase is included in Table 2.

To quantify the impact of LWP uncertainties not only for clouds above open ocean but also above sea ice, the prognostic LWP is both increased and reduced artificially by 50 % in two sets of simulations, going along with the according increase and decrease of $r_{\mathrm{eff}}$. $\Delta F^{\uparrow}$ changes above sea ice from $-35\,\mathrm{W\,m^{-2}}$ to $-45\,\mathrm{W\,m^{-2}}$ by the reduction of the LWP and to $-28\,\mathrm{W\,m^{-2}}$ by the increase of the LWP. This qualitatively matches the findings from Solomon et al. (2023) that the IFS produces too small LWP in the central Arctic. The bias $\Delta F^{\uparrow}$ above open ocean changes from $29\,\mathrm{W\,m^{-2}}$ in the reference case to $-5\,\mathrm{W\,m^{-2}}$ by

the reduction of the LWP and to $47\,\mathrm{W\,m^{-2}}$ by doubling the LWP. This indicated that the adjustments of the IFS need to be different above sea ice compared to open ocean. Above sea ice, the increase of the LWP and the implicated $r_{\mathrm{eff}}$ improves $\Delta F^{\uparrow}$ (Fig. 8a) and above open ocean, the decrease of the LWP and the implicated $r_{\mathrm{eff}}$ reduces $\Delta F^{\uparrow}$ (Fig. 8b). This direction of changes matches qualitatively the findings of Young et al. (2016) and Moser et al. (2023) with higher $N_{\mathrm{d}}$ above sea ice than above open ocean, assuming a linear relation between LWC and $N_{\mathrm{d}}$ as found by Leaitch et al. (2016) and Dionne et al. (2020).

Although the bias $\Delta F^{\uparrow}$ above open ocean is largely reduced, $\mathcal{H}^{\uparrow}$ is increased to 0.39 due to significant changes in the shape of the irradiance distribution, especially at the upper end where the sharp cut-off of the highest irradiances is reduced from $470\,\mathrm{W\,m^{-2}}$ to $440\,\mathrm{W\,m^{-2}}$ because of the lower LWP.





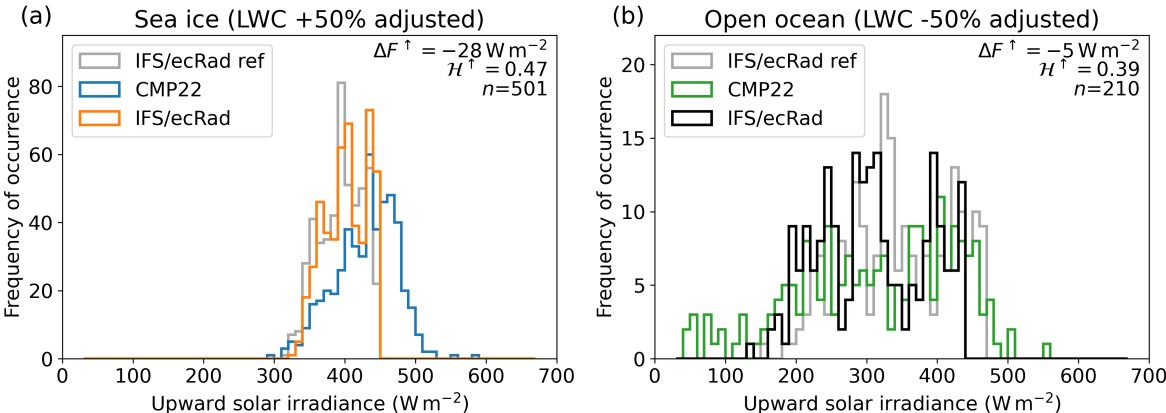

**Figure 8.** Distribution of upward solar irradiances measured by the CMP22 (blue) and simulated by ecRad (a) above sea ice with a 50 % increased LWP and (b) above open ocean with a 50 % decreased LWP with subsequent adjustments to the liquid effective radius (orange) together with the reference simulations (grey).

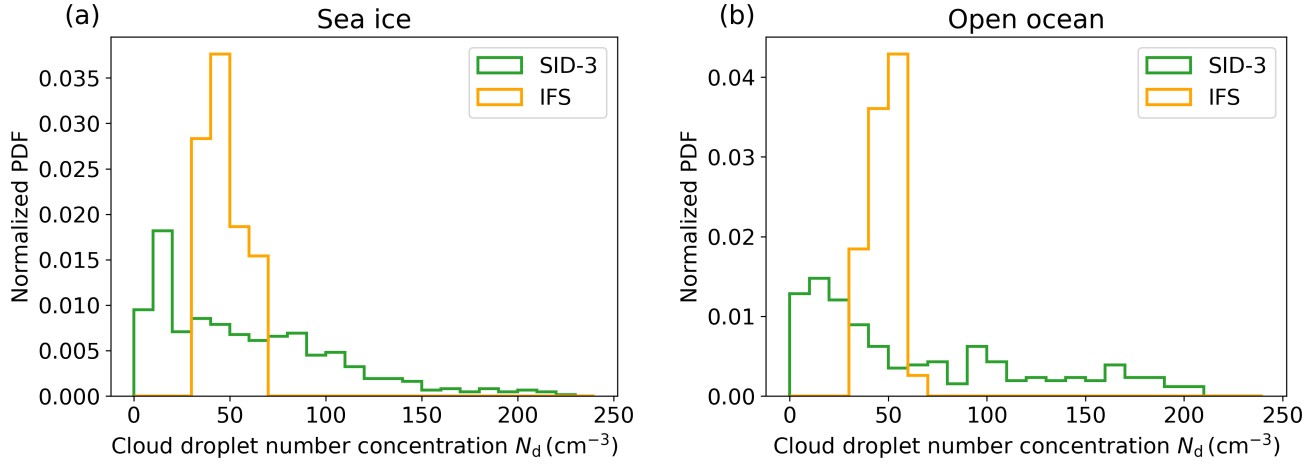

**Figure 9.** Distributions of cloud droplet number concentrations measured by the SID-3 and parameterized within IFS above (a) sea ice and (b) open ocean averaged over 60 s intervals.

### 4.3.3 Cloud droplet number concentration

The cloud droplet number concentration affects the cloud radiative properties by occurring in Eq. 4. The parameterized $N_d$
in the IFS is compared to in situ observations available from the SID-3 cloud probe aboard Polar 6 (Schnaiter and Järvinen, 2019). However, the flight track of Polar 5 does not always match the path of Polar 6. Nevertheless on a statistical basis, Polar 5 and Polar 6 sampled the same cloud and air mass regimes. Figure 9 shows the result of the $N_d$ parameterization described at



the beginning of Sect. 4 together with the in situ observations for the filtered scenes. The number concentrations in the IFS are within a narrow range between $36\,\mathrm{cm^{-3}}$ and $69\,\mathrm{cm^{-3}}$ with slightly higher concentrations above sea ice due to slightly higher

prognostic wind speeds. The in situ observations show a much broader range up to $230\,\mathrm{cm^{-3}}$. The observed low values of $N_\mathrm{d}$ mostly result from cloud edges or cloud-free flight sections and are not comparable to the mean grid box values of the IFS. However, the high $N_\mathrm{d}$ of above $200\,\mathrm{cm^{-3}}$ measured by SID-3 are not captured by the IFS. The findings from Moser et al. (2023) for two different aircraft campaigns in the Arctic with higher $N_\mathrm{d}$ above sea ice compared to open ocean are different from the ACLOUD observations, which may be attributed to a different season and different dominating air masses.

To investigate the impact of more realistic cloud droplet number concentrations on the reflected solar irradiance, a new set of ecRad simulations is performed with adjusted $N_\mathrm{d}$. The lower boundary $N_\mathrm{d,obs,min}$ is fixed to $N_\mathrm{d,IFS,min} = 36\,\mathrm{cm^{-3}}$ to account for the IFS grid box size, which cannot resolve cloud edges with only a few cloud droplets. The upper boundary $N_\mathrm{d,obs,max}$ is set to $200\,\mathrm{cm^{-3}}$, excluding only the highest values of the distribution's tail. The initial $N_\mathrm{d,IFS}$ appearing as cloud droplet number concentration in Eq. 4 is replaced by

$$N'_\mathrm{d,IFS} = (N_\mathrm{d,IFS} - N_\mathrm{d,IFS,min}) \cdot \frac{N_\mathrm{d,obs,max} - N_\mathrm{d,obs,min}}{N_\mathrm{d,IFS,max} - N_\mathrm{d,IFS,min}} + N_\mathrm{d,obs,min}, \qquad (10)$$

where $N'_\mathrm{d,IFS}$ is the adjusted cloud droplet number concentration. $N_\mathrm{d,IFS,min}$ ($N_\mathrm{d,IFS,max}$) is the minimum (maximum) cloud droplet number concentration from the IFS parameterization and $N_\mathrm{d,obs,min}$ ($N_\mathrm{d,obs,max}$) the minimum (maximum) cloud droplet number concentrations derived from in situ observations. Figure 10 shows the result of these adjustments. In general, the increase of $N_\mathrm{d}$ increases the reflected solar irradiance. $\Delta F^\uparrow$ decreases by scaling $N_\mathrm{d}$ above sea ice to $-27\,\mathrm{W\,m^{-2}}$, but

increases above open ocean to $48\,\mathrm{W\,m^{-2}}$. $\mathcal{H}^\uparrow$ changed to 0.41 above sea ice and to 0.38 above open ocean accordingly. Above sea ice, the $N_\mathrm{d}$ parameterization may be optimized by a higher variability. Above open ocean, this larger variability of $N_\mathrm{d}$ increases the overestimation by ecRad. A minor issue are observed differences of $N_\mathrm{d}$ between sea ice and open ocean surface, which are not taken into account by the parameterization in IFS.

## 5   Conclusion

Airborne observations of broadband solar irradiance measured above Arctic low-level clouds during the ACLOUD airborne campaign in May/June 2017 were used to evaluate the corresponding solar irradiances simulated by the IFS of the ECMWF. For this purpose, the ecRad radiative transfer scheme embedded in IFS was run in an offline mode using the output of the corresponding IFS 00 UTC runs as input. While there is agreement within the observational uncertainty between the measured and simulated downward solar irradiance, larger differences exceeding the pyranometer's uncertainty are found for the upward

solar irradiance. In a sensitivity study constrained by surface and cloud properties observed during ACLOUD, this bias was attributed to issues of the IFS in representing the sea ice albedo and low-level, liquid-dominated mixed-phase clouds. The impacts of different surface and cloud properties were quantified. The limitations of the sea ice model by Ebert and Curry (1993) to represent the change of sea ice albedo during the melting season cause more than $50\,\%$ of the observed bias. A comparison to airborne observations reveals an underestimation of the sea ice albedo by the IFS, especially in the wavelength



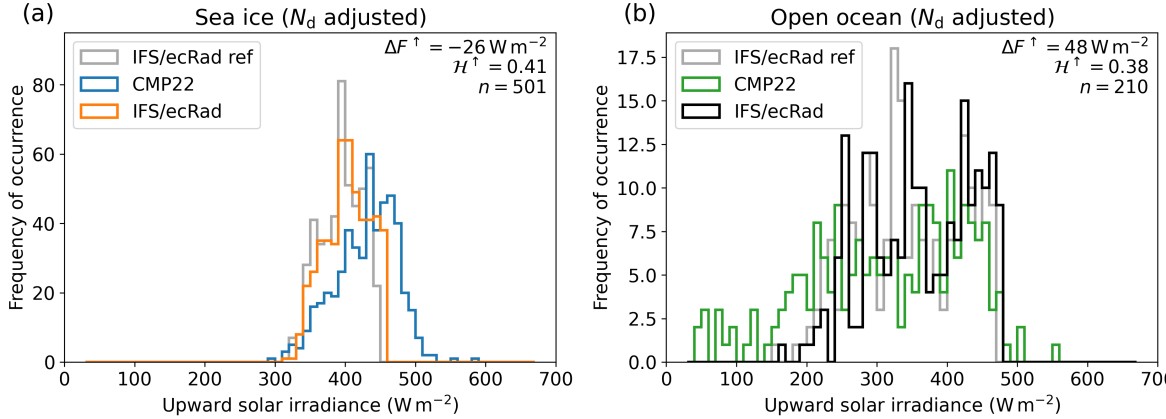

**Figure 10.** Distribution of upward solar irradiances above (a) sea ice measured by the CMP22 (blue) and simulated by ecRad with the adjusted number concentration and liquid effective radius (orange) together with the reference simulations (grey) and above (b) open ocean measured by the CMP22 (green) and simulated by ecRad with the adjusted number concentrations and liquid effective radius (black) together with the refence simulations (grey).

band from $690\,\mathrm{nm}$ to $1190\,\mathrm{nm}$. Implementing the measured sea ice albedo values into ecRad decreases the bias between the simulations and the observations to $-16\,\mathrm{W\,m^{-2}}$.

A misrepresentation of cloud fraction is assessed by active cloud remote sensing. The observed cloud fraction does not change $\Delta F^{\uparrow}$ above sea ice, but reduces the bias from $29\,\mathrm{W\,m^{-2}}$ to $18\,\mathrm{W\,m^{-2}}$ above open ocean where the observations show lower cloud fractions and the difference between the dark ocean and clouds is particularly large. The impact of cloud ice was quantified by artificially changing the IWP of the IFS output. The sensitivity of the upward solar irradiance to variations of the IWP of the underlying clouds is nearly negligible with the largest impact of $-2\,\mathrm{W\,m^{-2}}$ above open ocean by reducing the IWP by $50\,\%$. The cloud optical properties strongly depend on the LWP of the clouds. Confronting the prognostic LWP with airborne observations (above open ocean only), reduces the positive $\Delta F^{\uparrow}$ strongly and overcompensates it with a bias of $-28\,\mathrm{W\,m^{-2}}$. To estimate the effect of a misrepresentation of LWP also above sea ice, a non-observation based sensitivity study was performed. Increasing the LWP by $50\,\%$, $\Delta F^{\uparrow}$ improves above sea ice to $-28\,\mathrm{W\,m^{-2}}$ and by decreasing the LWP by $50\,\%$, $\Delta F^{\uparrow}$ improves to $-5\,\mathrm{W\,m^{-2}}$.

Airborne in situ observations have shown that the range of $N_d$ in the IFS is significantly smaller than measured. This affects the cloud radiative properties simulated by the IFS. Adjusting $N_d$, which occurs in the parameterization of the liquid effective radius from Martin et al. (1994) within a range of $36\text{-}69\,\mathrm{cm^{-3}}$, to a broader range of number concentrations found in the observations ($36\text{-}200\,\mathrm{cm^{-3}}$), results in a bias reduction above sea ice to $-27\,\mathrm{W\,m^{-2}}$ and in a bias enlargement to $48\,\mathrm{W\,m^{-2}}$ above open ocean.

The sensitivity study identifies the misrepresentation of the surface albedo as the largest contributor to the bias above sea ice. The sea ice albedo values in the IFS are applied as representative constant albedo values of dry snow, melting snow and bare sea



ice for fixed times of the year. Replacing these with a sea ice albedo parameterization that considers mixtures of different sea
ice types and their specific albedos depending on parameters like the surface temperature may improve the ability of the IFS in correctly simulating the upward solar irradiances in the Arctic (Jäkel et al., 2019b, 2023). The uncertainties of cloud radiative effects in the IFS significantly depend on the surface type below the clouds. With large contributions to the bias improvement given by realistic cloud droplet number concentrations and liquid water paths above sea ice and by realistic cloud fractions and liquid water paths above open ocean, a large amount of the bias could be attributed to the representation of cloud microphysical
properties.

*Data availability.* All ACLOUD-related data sets used in this study are publicly availabe on the PANGAEA Data Publisher (https://www.pangaea.de/) and referenced in Sect. 2.1. The IFS output used in this study was downloaded directly from the ECMWF servers using the Meteorological Archival and Retrieval System (registration required). Aerosol, $CH_4$, CO and $NO_2$ data were downloaded from the Copernicus Atmosphere Monitoring Service (CAMS) Atmosphere Data Store (ADS) on https://ads.atmosphere.copernicus.eu/cdsapp#!/dataset/
cams-global-reanalysis-eac4. $CO_2$ and $N_2O$ data were downloaded from the Copernicus Atmosphere Monitoring Service (CAMS) Atmosphere Data Store (ADS) on https://ads.atmosphere.copernicus.eu/cdsapp#!/dataset/cams-global-greenhouse-gas-inversion. This study contains modified Copernicus Atmosphere Monitoring Service information (2023); neither the European Commission nor ECMWF is responsible for any use that may be made of the Copernicus information or data it contains. Ozone soundings used in this study were performed by the Alfred-Wegener-Institute, Helmholtz Centre for Polar- and Marine Research; the ozone data used in this publication were obtained
from Peter von der Gathen as part of the Network for the Detection of Atmospheric Composition Change (NDACC) and are available through the NDACC website www.ndacc.org.

*Author contributions.* HM performed the analysis, the radiative transfer simulations and drafted the manuscript. AE and MW contributed to conception and design of the study. RH and JR contributed to the set up of ecRad. All authors contributed to the discussion of the results. All authors contributed to reviewing and editing of the manuscript.

*Competing interests.* The authors declare that they have no conflict of interest.

*Acknowledgements.* We gratefully acknowledge the funding by the Deutsche Forschungsgemeinschaft (DFG, German Research Foundation) – project number 268020496 – TRR 172, within the Transregional Collaborative Research Center "ArctiC Amplification: Climate Relevant Atmospheric and SurfaCe Processes, and Feedback Mechanisms (AC)[3]".



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
