# Peer review of "Evaluation of downward and upward solar irradiances simulated by the Integrated Forecasting System of ECMWF using airborne observations above Arctic low-level clouds"

_EGUsphere, 2023_

## Author Comment (AC1)

**Authors' response to reviewer #1 comments on the manuscript:**

**"Evaluation of downward and upward solar irradiances simulated by the Integrated Forecasting System of ECMWF using airborne observations above Arctic low-level clouds"**
**[EGUSPHERE-2023-2443]**

We thank the anonymous referee for his/her time and for the many suggestions and comments, which helped us to improve the manuscript. After repeating the reviewer's comments in black we detail our responses to each of these comments in blue below. The line numbers contained in the authors' response correspond to the revised version of the manuscript.

We have also taken the opportunity to

A) correct rounding errors of some differences between simulated and observed irradiance in the manuscript to reach consistency between the values indicated in the different figures, the tabular overview and the text passages, and

B) rename "LWC" to "LWP" in Fig. 7 and Fig. 8 to reach consistency within the manuscript.

**Summary:** This paper provides an overview of comparisons between aircraft-based radiation measurements and output from the IFS global model. The paper provides some interesting perspectives, though the overall ability to draw firm conclusions is limited by the sample size connected to this comparison. Therefore, it is challenging to say that there are any significant break throughs that come because of this work, though the conclusions provide interesting perspectives on potential avenues for model improvement. I believe that this work is suitable for ACP and that it should be considered for acceptance after some minor items have been accounted for. These are outlined below.

**Major Comments:**

Lines 233-235: I agree that the low values in the measurements are due in part to the gradient between dark ocean and bright clouds, but isn't it also simply due to the broader range of downwelling irradiances sampled during these times with higher solar zenith angles?

➢ We agree that the broader range of downward irradiances, which is striking when comparing Fig. 2b to Fig. 2a (from the manuscript), plays an additional role for the low values of the CMP22 measurements. Figure 1 (from this authors' response) displays the measured solar irradiances above open ocean together with the solar zenith angle for both the downward (black) and upward (orange) direction. Below 150 W m$^{-2}$ the solar irradiances correspond to a mixture of low and high solar zenith angles, below 100 W m$^{-2}$ small solar zenith angles from around noon are dominating (clustering around 58°).

➢ We included the influence of the solar zenith angles in the justification for the low values, which now reads :
"The low values of the measurements result from a combination of higher solar zenith angles, which are frequently present above open ocean, and scenes without any clouds below Polar 5 where the dark open ocean absorbs the major part of the incoming solar radiation." (lines 232-234)

[Figure]

Figure 1: Solar irradiance measured by CMP22 above open ocean together with the solar zenith angles for downward (black) and upward (orange) direction

Section 4.2: It seems likely that some of the differences between model output and aircraft measurements are the direct result of spatial variability in the 2D plane — particularly the case when CF ranges from 0-1 over open water. The aircraft is only taking a single transect, which could readily see differences between 0-100% when the grid-box average is around 60%. Please consider adding some discussion or sensitivity studies related to the 2D sampling volume that is being captured by a single line from the aircraft data.

> The analysis reveals a sub-grid variability of the observed cloud fraction that cannot be seen by the IFS, where the cloud fraction is a grid box mean. However, this imbalance is partly mitigated by applying a 60 s average to the remote sensing based cloud fraction to mimic the grid box size. To fully tackle the imbalance, a flight pattern would have been needed where whole grid boxes are systematically sampled, but such patterns were not performed during the ACLOUD campaign. Nevertheless, if a flight path during a cold air outbreak with developed streets of roll convection above open ocean lies in between two adjacent cloud streets, a certain imbalance in cloud fraction may be caused. Anticipating Fig. 3 (further below in this reply), which shows an example of LWP when flying perpendicular to cloud streets from a cold air outbreak, reveals at least for the LWP that it rarely decreases to 0 g m$^{-2}$ between adjacent cloud streets and it may subsequently apply for a cloud fraction of 0 as well.

> The following is added to the manuscript:
> "This cloud fraction is calculated from 60 s flight sections. As these flight sections do cover only a cross section of a grid box, there is an imbalance between observed variability of cloud fraction and the grid box mean. Especially low cloud fractions are less likely in IFS." (lines 296-298)

Line 301-302: This conclusion incorporates some assumptions about sun angles and the relative albedos of clouds and underlying ice. Some discussion is likely warranted.

> We agree that differences depend also on other parameters and our conclusion cannot be generalized. This becomes obvious by some outliers not strictly following our conclusion. However, from theory and radiative transfer simulations it can be concluded that the sign of the differences depends on under-/overestimation of cloud fraction. A higher cloud fraction will always increase upward irradiance. This holds also for clouds over sea ice, as they reduce water vapor absorption. The outliers in Fig. 4 (manuscript) are likely caused by the uncertainty of cloud fraction comparing the entire grid box with the cloud fraction along the aircraft track.

> ➤ The manuscript now reads as follows:
> "In theory, the data above the 1:1 line in Fig. 4b correspond to an underestimation of the cloud fraction by the IFS, coming along with negative irradiance differences indicated by orange colors, and vice versa. Data that do not follow this theory are likely caused by the possible imbalance of cloud fractions induced by comparing the entire grid box with the aircraft transect." (lines 305-308)

Section 4.3.1: I actually consider ice water path to be relatively "macro" physical in nature. What if the IWP is fine, but the crystal shapes and sizes are incorrect? How would this impact the radiation? Please consider adding some discussion.

> ➤ We agree and renamed Section 4.3 to "Macro- and microphysical cloud properties" to cover both sensitivity studies with IWP/LWP and cloud droplet number concentration. The effect of ice crystal shape and size for Arctic mixed-phase clouds becomes relevant only when IWP is significantly large and at least in the same range as LWP (Ehrlich et al., 2008). For ACLOUD, most clouds were dominated by liquid droplets (Klingebiel et al., 2023) so that these effects can be neglected as a first assumption. However, simulations testing the IFS-operational ice optics parameterization from Fu (1996) against the experimental ice optics parameterization from Yi et al. (2013) are displayed in Fig. 2 (in this reply) for one exemplary ACLOUD flight. It shows the differences between Yi and Fu ice optics in the upward irradiance at 2.8 km altitude. The bias in the area of interest covered by the campaign is below 1 W m$^{-2}$ and reaches outside the campaign area up to 5 Wm$^{-2}$ when the IWC in the IFS is particularly high. However, such an analysis is not applicable to a model-observations-comparison due to an unclear agreement between the ice water content in the IFS clouds and the observed clouds.
> ➤ We decided to add the following paragraph for a short discussion to the manuscript:
> "Similarly, ice crystal shape and size will not significantly impact the irradiance reflected by these liquid dominated clouds (Ehrlich et al., 2008). Also the choice of the ice optics parameterization within ecRad can impact the reflected irradiance (as shown by e.g. Wolf et al. (2020)). However, a model-observations-comparison in this regard would require an agreement between the IWC in the IFS and the observations. This agreement cannot be verified with the Polar 5 instrumentation." (lines 338-342)

[Figure]

*Figure 2: Differences of upward solar irradiance at 2.8 km altitude between ecRad runs with the ice optics parameterization from Fu (1996) and Yi et al. (2013).*

Paragraph starting at line 330: This paragraph can be clarified a bit. If I understand correctly, there are two issues at hand, namely: 1) the fact that the LWP observation is only available above open ocean; and 2) The fact that the IFS does not provide an LWP quantity that can be readily compared, due to the fact that some clouds may be above the aircraft. I would expand upon what was done a little to make it clear. At the moment, it sounds like the integration of IFS LWC is meant to solve the "over-ocean-only" problem in the observations. It took me three read-throughs to fully understand that this is in fact two separate issues.

> Thanks for pointing this out, the confusion was not our intention. We solved it with a clear separation between the two mentioned issues and rephrasing the paragraph.
> The manuscript now reads:
> "Above sea ice with its high emissivity the retrieval sensitivity is not sufficient, which is why this sensitivity study is firstly limited to open ocean. To confront the observed LWP above open ocean with the IFS output, the prognostic liquid cloud mass mixing ratio is converted to LWC and vertically integrated between the surface and Polar 5 flight altitude to the IFS LWP." (line 346-349)

Paragraph starting on line 335: I would like to see more information on how these LWP values were calculated. For example, is this only for "cloudy-sky" times? Or is this the integrated average over the 60 s window? What about in the model? Is this the "all sky" LWP? Or the cloud LWP? Assuming that it's a grid-box average for cloudy conditions only, do you have information on the variability within each grid box? To what extent can some of this simply be due to sampling across a 2D cloud field?

> The LWP from ACLOUD and from IFS in this analysis is based on all times with available retrieval by MiRAC, including both cloud-free and cloudy conditions ("all-sky" LWP). Nevertheless, the main aim of ACLOUD was to measure Arctic clouds, which is why cloud-free times are in the minority. The measured LWP from Polar 5 was integrated over the mentioned 60 s window to mimic the grid box size behind the IFS LWP. The IFS LWP is a grid box mean and not the in-cloud LWP. Regarding horizontal variability of the LWP, the standard deviation of the in-cloud liquid water content equals the in-cloud mean water content in the operational IFS configuration. To estimate the effect of sub-grid LWP variability, an extreme case from ACLOUD is analyzed. Figure 3 (in this reply) shows an example of LWP measured during a flight section crossing typical roll convection perpendicular to the rolls during a cold air outbreak. By calculating the 60 s averages (orange data points), most of the LWP variability due to the roll convection is eliminated. This suggests that the airborne observations are comparable to the grid box mean. For roll convection of larger amplitude this might differ (see also reply to second major comment).
> The manuscript now reads:
> "The combinations of the observed and the prognostic LWP are shown for the 210 scenes in Fig. 6. Observations are again 60 s averages of LWP including cloud-free data and IFS values are the grid box mean all-sky LWP. They reveal the tendency of the IFS to predict too high LWP, while the observations rarely show a LWP above 150 g m$^{-2}$ [...]. Similar to the discussion in Sect. 4.2, this mismatch partly results from the imbalance of Polar 5 sampling along a straight flight leg and IFS providing grid box means." (lines 350-355)

[Figure]

*Figure 3: Variability of high-frequency retrieved LWP and 60 s averages with standard deviation during an exemplary cold air outbreak with developed cloud streets*

Line 360-361: "This indicates that the adjustments of the IFS need to be different above sea ice compared to open ocean": This is speculative and based on a very limited dataset that was collected under a very limited number of seasons and synoptic regimes. Also, what would be the physical explanation for this need of different adjustments? Please consider adding more discussion here.

> We agree that this phrase is too speculative without mentioning the limitation to the ACLOUD dataset. Regarding physical explanatory approaches: One reason simply results from radiative transfer accounting for the different surface albedo of sea ice and open ocean. Due to the high reflectivity over sea ice, an increase of LWP changes the above cloud reflectivity less than the same increase of LWP over open ocean. Furthermore, clouds over sea ice are known to be different to clouds over open ocean for several reasons. Above ocean, especially during cold air outbreaks, turbulent surface fluxes of sensible and latent heat are magnitudes larger than above sea ice (Hartmann et al., 1997). Aerosol concentrations (and subsequently cloud droplet number concentrations) are known to differ as well (Young et al., 2016; Moser et al., 2023). These reasons could have lead to a better or worse representation of the clouds in IFS above open ocean.

> The manuscript is changed as follows:
> - To the initial "This indicated that the adjustments of the IFS need to be different above sea ice compared to open ocean" the limitation "to match the observations during the ACLOUD campaign" is appended. (line 379)
> - After describing the improvement by different directions this short discussion is added:
> "Possible reasons for the different LWP change directions may lie in differences of cloud physics between sea ice and open ocean. Above open ocean, especially during cold air outbreaks, turbulent surface fluxes of sensible and latent heat are magnitudes larger than above sea ice (Hartmann et al., 1997) and cloud droplet number concentrations are known to differ (Young et al., 2016)." (lines 383-386)
> - "This direction of changes matches qualitatively the findings of Young et al. (2016) and Moser et al. (2023) […]" is replaced with "The changes shown by the sensitivity study match qualitatively the findings of Moser et al. (2023) […]". (line 387)

Lines 361-362: The authors discuss "improving" DeltaF and then discuss "reducing" DeltaF. It's not clear to the reader how to interpret "improve" versus "reduces". Please stick with one theme (e.g., improves and worsens, or increases and decreases).

> The word "reduces" was replaced with "improves" (line 382). The confusion in the initial manuscript was introduced by avoiding the wording "increase $\Delta F^{\uparrow}$" for the less negative $\Delta F^{\uparrow}$, which is an improvement in the case with negative $\Delta F^{\uparrow}$ above sea ice. Thanks to this comment, we reached consistency now.

Line 374: It is not clear to the reader why or how droplet concentrations and wind speeds are linked.

> The cloud droplet number concentration is parameterized in IFS as a function of aerosol number concentration just below cloud base, which in turn is parameterized as a function of near surface aerosol mass concentrations (following Martin et al. (1994), Boucher and Lohmann (1995) and Lowenthal et al. (2004)). This, in turn, is parameterized in dependence of the prognostic surface wind speed following Erickson et al. (1986) and Genthon (1992), taking the injection of sea spray aerosols into account. At the end, the increase of the wind speed translates to an increasing number concentration (up to a specific wind speed). This relation was briefly mentioned earlier (after introducing Eq. (4)). We decided to add here "…leading to a stronger injection of sea spray aerosols to the atmosphere, which can act as cloud condensation nuclei." to clarify this relation for the reader. (lines 399-400)

Line 375: Are the in-situ cloud drop distribution data averaged over a grid box as well (~60 s)? Or are these distributions of high-resolution data?

> To enable a fair comparison, the in situ cloud droplet number concentration measurements are averaged over 60 s intervals as well. This is already mentioned in the caption of Fig. 9 and for clarification we decided to include this information in line 397 by adding "averaged over 60 s intervals".

Lines 392-393: Are the physical mechanisms supporting differences between sea ice and open ocean clouds described in a separate publication? If so, please include a reference. If not, please provide some background on what drives these differences.

> One main source of sea spray aerosols in the Arctic is wind-induced breaking of ocean waves (Nilsson et al., 2001), which can obviously only occur at the interface between ocean and atmosphere. Sea spray aerosol production above (snow-covered) sea ice happens as well, but by sublimating saline snow particles during blowing snow events (Yang et al., 2019). As a recent exemplary reference, Confer et al. (2023) is added to the manuscript, covering both mechanisms above sea ice and open ocean, in the following way:
> "[…] but can occur by different sea salt aerosol production mechanisms above sea ice (blowing snow) and open ocean (wave breaking), as described by e.g. Confer et al. (2023)." (lines 422-423)

Lines 424-426: Regarding a parameterization that accounts for different sea ice types, It would have been interesting to evaluate the output of such a parameterization in the context of these runs. It seems like all of the inputs would be available, and then the outputs could be inserted into the offline simulations to assess this specific claim.

> We agree that implementing a more sophisticated sea ice albedo parameterization considering different sea ice types would be a significant improvement of IFS. However, applying such a parameterization is not possible in the context of these runs, as the IFS does

> not provide the fractions of sea ice subtypes (bare sea ice, snow-covered sea ice, melt ponds). These are required for implementing the parameterization of Jäkel et al. (2019).

Line 429-430: Cloud macro- and microphysical properties? Also, again please keep in mind that you have only a small sample size from which to evaluate the performance of the IFS cloud parameterizations.

> ➢ We replaced "cloud microphysical properties" by "cloud micro- and macrophysical properties" (lines 459-460). Regarding the small sample size, the constraint "during the ACLOUD campaign" is added in line 462.

**Minor Comments:**

The authors sometimes use "the" excessively. For example, in lines 28-34 all of the bolded "the" occurences can be removed, in my opinion: "One obvious issue in the Arctic results from **the** sparse observational coverage, which limits **the** data assimilation (Bauer et al., 2016; Jung and Matsueda, 2016; Lawrence et al., 2019; Ortega et al., 2022). Furthermore, **the** modeling of the sea ice cover is a major obstacle for correctly representing the Arctic surface energy budget but is still uncertain due to the complexity of sea ice dynamics (Day et al., 2022). **The** representation of low-level Arctic clouds and especially mixed-phase clouds has been identified as another major source of uncertainty (Forbes and Ahlgrimm, 2014). As shown by Morrison and Pinto (2006), especially the cloud microphysical schemes cause uncertainties in **the** cloud phase and precipitation." I recommend that the authors re-read the paper and evaluate their use of this short word.

> ➢ Thanks for this comment, we reevaluated the use of this short word throughout the entire manuscript.

Line 157: "no prognostic variables" should be "not prognostic variables".

> ➢ The word "no" was changed to "not" (line 155).

Line 228: Recommend changing this to read "Observations of upwelling irradiance above sea ice surfaces…"

> ➢ This was rephrased to "Observations of upward irradiance above sea ice surfaces…" (line 227). We decided for the term "upward" instead of "upwelling" to be consistent both within the manuscript and with the WMO terminology.

Figure 6: It could just be my eyes playing tricks on me, but when I do some quick counting on the dots in this figure and the associated "occurrences", my intuition says this is more than the 210 cases over water that were mentioned above. Can you confirm that the totals represented by these dots sum up to 210?

> ➢ Well spotted. The associated occurrences summed up to 743 cases and not to the expected 210 cases. These 743 cases correspond to timesteps with an available LWP retrieval, but this information was missing. The expected 210 cases are composed of the overlap between the 743 cases with available LWP retrieval, 886 cases of cirrus-free conditions and 615 cases of open ocean conditions. Figure 6 (in the manuscript) is replaced with a version containing the 210 cases only (Fig. 4 in this reply). To clarify that this figure relates to the 210 scenes, the words "for the 210 scenes" are added in line 350.

[revised manuscript text omitted]

---

## Author Comment (AC2)

**Authors' response to reviewer #2 comments on the manuscript:**

**"Evaluation of downward and upward solar irradiances simulated by the Integrated Forecasting System of ECMWF using airborne observations above Arctic low-level clouds"**
**[EGUSPHERE-2023-2443]**

We thank the anonymous referee for his/her time and for the helpful comments and suggestions, especially on a better description of how the sensitivity study was performed, which helped us to improve the manuscript. After repeating the reviewer's comments in black we detail our responses to each of these comments in blue below. The line numbers contained in the authors' response correspond to the revised version of the manuscript.

We have also taken the opportunity to

A) correct rounding errors of some differences between simulated and observed irradiance in the manuscript to reach consistency between the values indicated in the different figures, the tabular overview and the text passages, and

B) rename "LWC" to "LWP" in Fig. 7 and Fig. 8 to reach consistency within the manuscript.

The authors use ACLOUD field campaign data to quantify the representation and uncertainty of Arctic low-level clouds in the ECMWF Integrated Forecasting System (IFS). The model's horizontal resolution is 1.4 km to 7.8 km. It has 137 vertical levels. To focus on low-level clouds, the authors only use cases with clouds below the level of Polar 5 and no cloud above. The comparisons of IFS and observed downward and upward irradiance are presented statistically to avoid the error due to temporal and spatial mismatch. To quantify the sensitivity of cloud properties, cloud fraction, ice water path, liquid water path, and cloud particle number concentration, as well as surface albedo are perturbed. Then a off-line radiative transfer code is used to assess the impact on the upward irradiance. The authors find the sea ice spectral albedo is the largest contributor to the bias in the upward irradiance above sea ice.

The analysis demonstrates the utility of field campaign data to evaluate high resolution models. This paper can potentially become a high impact paper. However, the use of statistical analysis in identifying cloud properties that are responsible for the bias is limited. In the end, the authors only use mean differences to evaluate the contribution. In addition, how the sensitivity study was performed needs to be clarified.

**Major issues**

Equations 3 and 4 show relationship among cloud properties. When the liquid water path is perturbed by keeping the shape of vertical profile, the liquid water content is scaled by the ratio of liquid water paths. The scaling liquid water content affects effective radius, according to Equation 4. It is not clear to me, therefore, when LWP is perturbed, whether this is a partial derivative or other cloud properties change according to Equations 3 and 4. This probably affect the result of the number concentration perturbation most. The authors need to describe more to clarify how the sensitivity study was performed.

> The scaling of specific parameters for the sensitivity study is not performed via a partial derivative, but with subsequent changes of the liquid effective radius triggered by initial adjustments of LWC and cloud droplet number concentration according to Eq. 4. For LWC adjustment (Sect. 4.3.2), the cloud droplet number concentration remains fixed. For cloud droplet number concentration adjustment (Sect. 4.3.3), the LWC remains fixed. The scaling of

IWC (Sect. 4.3.1) is followed by a subsequent adjustment of the ice effective radius according to Sun and Rikus (1999) and Sun (2001). In each relevant subsection, we added adequate details to clarify how the sensitivity study was performed for the relevant parameter:

- Section 4.3.1: "This increase (reduction) of IWP is propagated to an increase (reduction) of the ice effective radius according to Sun and Rikus (1999) and Sun (2001)." (lines 332-333)
- Section 4.3.2 (observation based): "The liquid effective radius is recalculated respectively, considering the changed LWC in Eq. 4. After adjusting, $r_{eff}$ is limited to 4-30 µm to match the IFS constraints again. $N_d$ remains fixed." (lines 362-364)
- Section 4.3.2 (non-observation based): "[…] going along with the according increase and decrease of $r_{eff}$. $N_d$ remains fixed." (lines 374-375)
- Section 4.3.3: "The liquid effective radius is recalculated respectively, considering the changed $N_d$ in Eq. 4, while LWC remains fixed." (lines 413-414)

The authors show distributions of irradiances in Figures. Also, Hellinger distances were computed. However, the authors do not discuss distribution differences. I would like to see discussions of distribution differences and how the distribution differences are used to narrow the uncertainty to identify cloud properties contributing the bias of upward irradiances.

➢ While Section 3 covers a discussion of multiple differences between various distributions in Fig. 2, we added some missing discussion of distribution differences and how they contributed to $\Delta F^{\uparrow}$ throughout the whole Section 4:

- Section 4.1 now contains "In comparison to the reference distribution, the adjusted distribution emerges with 320 W m$^{-2}$ at a 10 W m$^{-2}$ higher upward irradiance, but ends 20 W m$^{-2}$ higher at 470 W m$^{-2}$. The distribution itself is shifted to higher values throughout all upward irradiances with the highest peak located between 420-430 W m$^{-2}$." (lines 285-287)
- Section 4.2 now contains: "Above open ocean, occurrences of upward solar irradiance between 200 W m$^{-2}$ and 460 W m$^{-2}$ are mainly lower compared to the reference distribution. This reduction enables a new mode occurring at very low upward irradiances between 40 W m$^{-2}$ and 50 W m$^{-2}$. The replacement by the observed cloud fraction results in a higher amount of data with low reflected irradiances." (lines 320-322)
- Section 4.3.2 (observation based) now contains: "Compared to the reference distribution, the adjusted distribution emerges already at 80 W m$^{-2}$ instead of 150 W m$^{-2}$ and ends at upward irradiances 20 W m$^{-2}$ lower than before. The main mode ranges between 230 W m$^{-2}$ and 290 W m$^{-2}$ instead of between 320 W m$^{-2}$ and 340 W m$^{-2}$. Adjusting the LWP based on observations leads to a change in the correct direction by reducing the upward solar irradiances. Compared to the observations, this impact is too strong resulting in a conversion of the overestimation to an underestimation." (lines 366-370)
- Section 4.3.2 (non-observation based) now contains: "Above sea ice, the increase of the LWP and the implicit change of $r_{eff}$ improves $\Delta F^{\uparrow}$ (Fig. 8a) by a slightly higher emergence of the upward irradiance distribution at 320 W m$^{-2}$, by slightly higher irradiances over a wide range of the distribution and by keeping the same end of the distribution as in the reference case. Above open ocean, the decrease of the LWP and the implied $r_{eff}$ improves $\Delta F^{\uparrow}$ (Fig. 8b) by shifting the entire distribution to 20-30 W m$^{-2}$ lower upward irradiances." (lines 380-383)

- Section 4.3.3 now contains: "In general, the increase of $N_d$ increases the reflected solar irradiance. Above sea ice, the maximum values of upward solar irradiance reach 460 W m$^{-2}$ instead of 450 W m$^{-2}$ while the minimum remains unchanged. In between, the distribution is shifted to slightly higher irradiances. Above open ocean, the entire distribution of adjusted upward solar irradiances is shifted to higher irradiances, with the minimum and maximum ranging 10 W m$^{-2}$ higher." (lines 414-418)

**Minor issues**

Figure 2: dotted lines used for open ocean make the plot hard to see. Could them change to solid lines? Also tick labels of x-axis for the upper plots are missing. Are these the same as upward irradiance? Also, including mean irradiance values to the upper left helps.

> We changed in Fig. 2 the dashed lines above open ocean to solid lines. The x-axes of the downward irradiances is shared with the x-axes of the upward irradiances. For clarification the (shared) tick labels are now repeated at the ticks of the two upper panels. As suggested, the mean irradiance values are displayed at the upper left of each panel.

Figure 3: Are IFS sea ice albedo climatologies indicated by the solid lines?

> Exactly, the IFS sea ice albedo climatology from Ebert and Curry (1993) is indicated by the solid lines, as mentioned in the legend of Fig. 3a. To clarify this, the descriptions "circles" and "solid lines" were added to the figure caption, which now reads:
> "Time series of modes of measured sea ice albedo in the 400-690 nm, 690-1190 nm and 1190-2155 nm band (circles) and IFS sea ice albedo climatology (solid lines). The dashed lines show the parameterization for the measurements. […]"

**References**

Ebert, E. E. and Curry, J. A.: An intermediate one-dimensional thermodynamic sea ice model for investigating ice-atmosphere interactions, J. Geophys. Res., 98, 10085, https://doi.org/10.1029/93JC00656, 1993.

Sun, Z.: Reply to comments by Greg M. McFarquhar on 'Parametrization of effective sizes of cirrus-cloud particles and its verification against observations'. (October B, 1999,125, 3037–3055), Q.J Royal Met. Soc., 127, 267–271, https://doi.org/10.1002/qj.49712757116, available at: https://rmets.onlinelibrary.wiley.com/doi/10.1002/qj.49712757116, 2001.

Sun, Z. and Rikus, L.: Parametrization of effective sizes of cirrus-cloud particles and its verification against observations, Q.J.R. Meteorol. Soc., 125, 3037–3055, https://doi.org/10.1002/qj.49712556012, 1999.